# Effect of Direct Steam Injection and Instantaneous Ultra-High-Temperature (DSI-IUHT) Sterilization on the Physicochemical Quality and Volatile Flavor Components of Milk

**DOI:** 10.3390/molecules28083543

**Published:** 2023-04-17

**Authors:** Hao Ding, Zhaosheng Han, Bei Wang, Yadong Wang, Yawen Ran, Liebing Zhang, Yan Li, Chun Lu, Xiaoli Lu, Ligang Ma

**Affiliations:** 1School of Food and Health, Beijing Technology and Business University, Beijing 100048, China; 2College of Food Science and Nutritional Engineering, China Agricultural University, Beijing 100083, China; 3Junlebao Dairy Group Co., Ltd., Shijiazhuang 050221, China

**Keywords:** DSI-IUHT, physical properties, SPME, SAFE, volatile flavor

## Abstract

The effects of variations in the heat treatment process of milk on its quality and flavor are inevitable. This study investigated the effect of direct steam injection and instantaneous ultra-high-temperature (DSI-IUHT, 143 °C, 1–2 s) sterilization on the physicochemical properties, whey protein denaturation (WPD) rate, and volatile compounds (VCs) of milk. The experiment compared raw milk as a control with high-temperature short-time (HTST, 75 °C 15 s and 85 °C 15 s) pasteurization and indirect ultra-high-temperature (IND-UHT, 143 °C, 3–4 s) sterilization. The results showed no significant differences (*p >* 0.05) in physical stability between milk samples with different heat treatments. The DSI-IUHT and IND-UHT milks presented smaller particle sizes (*p <* 0.05) and more concentrated distributions than the HTST milk. The apparent viscosity of the DSI-IUHT milk was significantly higher than the other samples (*p <* 0.05) and is consistent with the microrheological results. The WPD of DSI-IUHT milk was 27.52% lower than that of IND-UHT milk. Solid-phase microextraction (SPME) and solvent-assisted flavor evaporation (SAFE) were combined with the WPD rates to analyze the VCs, which were positively correlated with ketones, acids, and esters and negatively associated with alcohols, heterocycles, sulfur, and aldehydes. The DSI-IUHT samples exhibited a higher similarity to raw and HTST milk than the IND-UHT samples. In summary, DSI-IUHT was more successful in preserving the milk’s quality due to its milder sterilization conditions compared to IND-UHT. This study provides excellent reference data for the application of DSI-IUHT treatment in milk processing.

## 1. Introduction

As a complex colloid, milk is heat-treated to extend its shelf life and reduce the likelihood of microorganisms and enzymes rendering the product unsafe or causing spoilage. However, thermal treatment can affect milk’s physicochemical and organoleptic qualities [1]. The most common heat treatments for milk include high-temperature short-time (HTST) and ultra-high-temperature (UHT) treatments. Heat treatment alters the salt and casein colloids in the milk, while the dissolved phases of calcium and phosphorus are transformed into the colloidal phase and the free calcium levels are reduced. Furthermore, some of the modifications caused by heating may lead to protein changes (directly manifested as whey protein denaturation (WPD) and its reaction with casein), altering the properties of colloidal casein. Compared to HTST, UHT-treated milk proteins are more prone to precipitation or gel formation. Prolonged storage causes fat uplift, increasing aggregation of fat globules and the formation of Maillard products which affects the stability of the milk [2,3]. In terms of flavor characteristics, dairy products contain a wide variety of volatile flavor substances. However, their levels are low, while the traits readily affected by heat treatment complicate the changes in the milk flavor substances [4]. The quality of raw milk influences the subsequent product and is related not only to microbial count but also to somatic cell count [5]. Studies have shown that consumers can distinguish between HTST- and UHT-treated milk, with most preferring the HTST product. Since the treatment conditions for HTST milk are milder, the flavor is closer to that of raw milk. UHT milk tends to present a cooked taste [6] due to the oxidation of the sulfhydryl groups (-SH) produced via β-lactoglobulin fission, resulting in sulfur dioxide production and possible adverse effects on flavor [7].

Milk is rich in protein, which is considered an important component contributing to flavor loss or release. Flavors can bind to proteins at low temperatures [8]. Temperature appears to affect binding properties, especially during thermal denaturation, where heat treatment induces protein unfolding. This modifies the protein structure, which decreases the binding constant and increases the number of binding sites [9]. Whey proteins (WP) are more susceptible to thermal denaturation than casein. In most cases, the interaction between proteins and flavor compounds is reversible and includes hydrophobic and hydrogen bonds [10]. In addition, the primary mechanism of interaction was believed to be hydrophobic interactions. κ-casein exhibits similar amphiphilic properties, while the other caseins are prevalently of hydrophobic natures. So, κ-casein plays the role of colloidal protector for all other caseins, making it possible for micelles to form. The caseins with polypeptide chains are folded into non-polar portions buried inside micelles. The unique structure of α-lactalbumin in WP may inhibit the exposure of hydrophobic regions, resulting in a low binding capacity to flavor compounds and a reduced ability to interact with flavor substances [11]. Bovine serum albumin (BSA) is the milk protein most capable of binding volatile flavor compounds, followed by β-lactoglobulin [12]. Studies have shown that β-lactoglobulin interacts with many flavor compounds, such as aldehydes, ketones, ionones, and esters. Tromelin and Guichard [13] applied 3D-QSAR molecular modeling for the first time to describe and predict flavor–β-lactoglobulin interaction, during which the bonding of β-lactoglobulin with hydrogen represents an important pathway. Furthermore, at least two β-lactoglobulin binding sites were confirmed [14]. Using UPLC-TOF-MS, Anantharamkrishnan et al. [15] discovered that β-lactoglobulin and 47 flavor compounds in 13 different functional groups in milk can form covalent bonds with most aldehydes, a part of sulfur, diacetyl, and allyl isothiocyanate. Moreover, alcohols can non-covalently interact with proteins [16]. The formation of covalent bonds (flavor proteins) may also be responsible for the loss or alteration of flavor.

The choice of time and temperature combinations for the heat treatment of milk has to consider both microbiological and product quality aspects to achieve optimal results. HTST reduces the impact on product quality but results in a shorter shelf life and requires low-temperature refrigeration for storage. However, the cold chain transportation systems in developing countries are inadequate to meet the demand for HTST milk. UHT processing mainly consists of indirect UHT and direct UHT (steam as a heating medium in direct contact with the product). It has a long shelf life and the convenience of room temperature transport, but the flavor produced by the longer periods of UHT sterilization currently used is unsatisfactory. The direct steam injection and instantaneous ultra-high-temperature (DSI-IUHT, 143 °C, 1–2 s) method is used as a direct sterilization technique of instantaneous UHT. The product enters the heating system in direct contact with the heating medium (steam). According to the conservation of volumetric flow, achieved by controlling the pressure and fluid velocity of the nozzle, the steam is better integrated with the product, and the residence time in the injection chamber is reduced. Then, flash evaporation cooling in the vacuum cylinder is performed, where the steam is removed and finally indirectly cooled to the packaging temperature. DSI-IUHT milk has a shelf life of three months when stored at room temperature, meeting the requirements of UHT sterilization, while instantaneous sterilization decreases the intensity of sterilization by reducing the heating time, reducing the destruction of proteins, and retaining the original milk flavor as much as possible. Compensating for the disadvantages of HTST and the current UHT sterilization technique, which none of the other UHT direct steam injection methods do, this study examines the texture and flavor of milk exposed to DSI-IUHT. It provides a theoretical basis for the production of milk with a long shelf life and good flavor in developing countries.

## 2. Results and Discussion

### 2.1. The Influence of Processing on the Physical Properties of Milk

#### 2.1.1. Particle Size and Physical Stability

The particle sizes of the different samples were characterized, and the results are shown in Figure 1.

The particle sizes of raw milk (3.41 ± 0.05 μm) were bigger among all samples, while the particle sizes decreased significantly (*p <* 0.05) as the sterilization intensity and the degree of homogenization increased. Similar particle sizes of 0.52 ± 0.02 μm and 0.54 ± 0.00 μm were observed in the DSI-IUHT and IND-UHT milks, respectively, with no significant differences (*p >* 0.05). However, substantial differences (*p <* 0.05) were apparent between the particle sizes of DSI-IUHT and HTST (75 °C and 85 °C) milk. The raw milk contained the largest particle size distribution, with sizes ranging from 0.296 to 27.183 µm, while the heat-treated milk mainly presented particles of approximately 0.296 µm to 4.05 µm. The particle size distribution was relatively similar between DSI-IUHT and IND-UHT. However, DSI-IUHT has a slightly smaller particle size distribution than HTST (75 °C and 85 °C). The particle sizes of all the samples were similar to those revealed by Wiking et al. [17], with an average fat globule size of about 4 μm (from 0.1 μm to 15 μm). The particle size distribution differences between the samples are related to homogenization and heat treatment, which are often combined in processing technology. As shown by the particle size distribution in Figure 1, the heat-treated samples have smaller particle sizes and a more concentrated particle size distribution than the raw milk. Due to the triple action of extrusion, strong impact, and loss of pressure expansion, the homogenizer destroys the milk fat globule membrane (MFGM) of the emulsion to form new, smaller lipoglobulin molecules [18,19] with larger surface areas and higher mobility [20]. Comparison of samples with primary homogenization (HTST, 75 °C 15 s and 85 °C 15 s) and samples with secondary homogenization (IND-UHT and DSI-IUHT) revealed significant differences in particle sizes and particle size distribution (*p <* 0.05), but there was no significant difference between samples with the same number of homogenizations (*p >* 0.05), indicating that homogenization changes the state of fat globules more than heat treatment.

Figure 2 shows the instability indices, where a higher instability index indicates a lower stability [21].

The instability index of the raw milk was significantly higher than the heat-treated milk (*p <* 0.05), with no significant differences between the different heat-treated milk samples (*p >* 0.05) and higher physical stability. Figure A1 and Figure A2 show the original picture of Figure 2 (raw milk and DSI-IUHT), respectively, with near-infrared light illuminating the entire sample cell. The physical stability is reflected through the evolution of the integrated transmission-time profiles and the original transmission as a function of sample position [22]. DSI-IUHT exhibited fewer changes in the transmission profiles over space and time compared with raw milk, resulting in better physical stability.

The results indicated that homogenization leads to smaller lipoglobulin formation, which can be spread evenly and stabilizes the entire emulsion system. Heat treatment affected particle size by changing the organization of proteins. WP was denatured when the milk was heated for a short time, as β-lactoglobulin produced different configurations and aggregates. Casein micelles with negatively charged surfaces and MFGM were used to achieve an emulsion system with better stability through a balance between intermolecular interactions and electrostatic mutual exclusion [23,24]. Moreover, milk’s ability to withstand the high-temperature treatment without losing its stability is quite unique. Studies [25] have shown that fresh milk at its natural pH can be heated at 140 °C for more than 10 min before solidification, enabling the production of many sterilized milk products with long shelf lives.

#### 2.1.2. The Rheological and Apparent Viscosity

The Brownian motion of the particles in static conditions was continuously observed to obtain the droplet mean square displacement (MSD), which corresponded to the diffusion particle coefficient. As shown in Figure 3a1–a5, the curves of all the samples were nonlinear, indicating overall viscoelasticity and unaltered structural stability.

However, slight differences were evident between the microrheological behavior of the different samples. As illustrated in Figure 3a1, the MSD curve measured in the raw milk displayed a denser distribution, while the Brownian motion range was more fixed. Furthermore, the structural stability was more susceptible to change, confirming the lower physical stability in Figure 2. Figure 3b,c show the results obtained through dynamic light scattering to exhibit the viscoelastic properties of milk treated with different methods [26,27]. No significant changes were evident between the macroscopic viscosity index (MVI) and elasticity index (EI) values of the samples in different heat treatment conditions after reaching the maximum value, indicating a stable internal emulsion network structure. The MVI value of DSI-IUHT was significantly higher than the others, indicating that the unique DSI-IUHT heat treatment method allowed the steam to be better integrated with the product, changing the droplet–droplet aggregation state via electrostatic interaction and facilitating the formation of a new stable network of porous, dense chain polymers, which promoted water molecule binding [28], increased the milk’s viscosity, limited the Brownian motion of the particles. The maximum value of EI of milk treated with 75 °C HTST was slightly higher than the others, and that of the 85 °C HTST-treated milk showed an increase at a monitoring time of around 11,000 s. HTST treatment induces the unfolding of WP molecules, exposing more hydrophobic groups from the cores of globular proteins and increasing surface hydrophobicity [29]. This leads to an increased number of hydrophobic groups participating in heat-induced gelation, resulting in a higher EI value [30]. As the intensity of heat treatment increases, peptide chains are stretched and re-aggregated, where hydrophobic interactions once again dominate and lead to a decrease in surface hydrophobicity. This also explains why DSI-IUHT and IND-UHT exhibit maximum elasticity values comparable to raw milk.

As shown in Figure 3b, DSI-IUHT displayed the highest MVI, corresponding to the shear viscosity in Figure 4.

The viscosity obtained using DSI-IUHT was significantly higher than in the other samples (*p <* 0.05). DSI-IUHT is an instant direct sterilization method that involves the rapid injection of high-pressure steam into milk, which penetrates the milk sample quickly, rapidly raising its temperature and causing WP denaturation and the cross-linking of casein proteins [31]. The resulting cross-linking may be the main factor contributing to the observed increase in the milk’s apparent viscosity. In contrast, HTST and IND-UHT are indirect sterilization methods, which cause a significant decrease in the milk’s apparent viscosity (*p <* 0.05). In these methods, heating induces a partial denaturation of milk proteins, which leads to protein aggregation and promotes large molecule interactions [32], mainly with soluble components such as carbohydrates and salts, thereby causing a decrease in the milk’s apparent viscosity.

#### 2.1.3. The WPD Rate

As shown in Table 1, no significant differences were evident between the total protein (TP) and non-protein nitrogen (NPN) values of the samples (*p >* 0.05), while the undenatured whey protein (UWP) values exhibited substantial differences (*p <* 0.05).

Studies have shown that WPs are more susceptible to thermal denaturation due to their higher numbers of secondary and tertiary structures than casein [33]. The degree of WPD in the milk increased at higher heat treatment temperatures between 75 °C and 143 °C. These results corresponded with previous studies [34,35]. The unique DSI-IUHT treatment process yielded a 27.52% lower whey denaturation rate than IND-UHT. The degree of WPD in the different heat-treated milk samples is shown in Figure A3, showing lighter WP lines for IND-UHT. DSI-IUHT showed a lower denaturation than IND-UHT because of the reduced residence time of the milk in the sterilization chamber. Akkerman et al. [36] found that the heat-treatment’s temperature more significantly affected whey denaturation than the heat-treatment’s time. For UHT milk, we can reduce the effect on WPD by using the instantaneous UHT process.

### 2.2. Analysis of the Odor-Active Compounds in Milk

#### 2.2.1. Quantitation of the Odor-Active Compounds in Milk

The solid-phase microextraction (SPME) method collects volatile compounds (VCs) from the headspace of a sample and is suitable for extracting volatile and semi-volatile flavors, which play important roles in aroma perception but may not represent the full range of aroma characteristics. The solvent-assisted flavor evaporation (SAFE) method is an effective way to extract volatile components from complex matrices and to better preserve the original natural flavors of the sample. Since the heat-treated milk contained more types and lower levels of volatile substances, both these extraction techniques were used to identify 57 VCs from 7 classes in the 5 milk samples, including 16 aromatic heterocyclic compounds, 12 acids, 10 alcohols, 7 aldehydes, 7 esters, 3 sulfuric components, and 2 ketones (Table 2 and Table 3). Here, 29 and 47 compounds were detected via SPME and SAFE, respectively.

A total of 16 flavor compounds were detected in the aromatic and heterocyclic compounds with complex sources and unstable frequencies [37]. The SPME method (Table 2) detected nine compounds, among which benzaldehyde was only present in the IND-UHT and DSI-IUHT milk samples. Although benzaldehyde can cause a bitter almond taste, its content was lower, and its flavor had a lower impact. Limonene is mainly found in cattle feed. Although the SAFE method (Table 3) detected limonene in all five samples, the results were similar to those revealed by Contarini et al. [38]. Its content was significantly lower in the IND-UHT and DSI-IUHT samples than in the HTST milk (75 °C HTST, 85 °C HTST). With the increase in sterilization temperature, the limonene molecules may undergo structural changes or decomposition.

Acids are derived from triglyceride hydrolysis occurring in milk fat due to endogenous enzymes or physical conditions or from the degradation of lactose and amino acids [39,40]. A total of 12 acids were detected, including formic acid, acetic acid, butanoic acid, and short-chain fatty acids. Formic acid is an advanced Maillard reaction product, which is not found in raw milk. Furthermore, formic acid was primarily responsible for the decrease in pH [41,42]. A small amount of butanoic acid was present in the raw and HTST milks, representing fermentation products from the energy metabolism of the ruminants [43]. Compared with the SPME method, the detection rate of eight acids, including propanoic acid, heptanoic acid, and 9-decenoic acid, was higher when using the SAFE method. The flavor was considered an important supporting factor for the milky smell.

Although alcohols are typically products of aldehydes, their flavor threshold values are higher [36,44]. In terms of their content, the SAFE method (Table 3) will show higher total alcohol content in raw and HTST milk than IND-UHT and DSI-IUHT. Only the SAFE method detected 1-Tetradecanol in the raw milk and HTST, probably because the high temperature caused the loss or conversion of alcohol compounds to other substances. As a long-chain alcohol detected by both SPME (Table 2) and SAFE (Table 3), 1-dodecanol presented a floral aroma and was derived from microbial metabolism.

The aldehyde odor threshold was generally low; e.g., the SAFE dodecanal threshold (Table 3) was only 0.14 μg/kg [45]. Seven aldehydes were detected by the SAFE method, while only one was detected by the SPME method. Comparing the detection rates of aldehydes between the two methods, the SAFE method can better detect aldehydes. Aldehydes can be produced as a flavor precursor via Strecker degradation, amino acid transamination, and fatty acid metabolism [46]. The SAFE method indicated that the aldehyde species and content were relatively high, with hexanal and nonanal aldehydes found in all samples, producing grassy, fruity, and woody aromas in the milk [47]. These straight-chain aldehydes are lipid oxidation products. The aldehyde content decreased at a higher bactericidal intensity.

Esters are obtained via the esterification of hydrolyzed free fatty acids and short-chain fatty alcohols when exposed to endogenous esterase. Methyl or ethyl esters composed of C1-C20 fatty acids represent the main ester substances in milk [48]. Although ester compounds have high flavor threshold values, their precursors (alcohol) are ten times higher [49]. The SPME method (Table 2) detected that only IND-UHT milk has hexadecanoic acid and methyl ester contents (about 9.37 ± 0.16 μg/kg), which are usually expressed as creamy and lightly fruity. The SAFE method (Table 3) detected hexadecanoic acid and methyl ester in all five samples, while the content of these compounds decreased at a higher bactericidal intensity. Milder bactericidal conditions may be more conducive to the presence of hexadecanoic acid and methyl ester.

Sulfur compounds are responsible for the cooked flavor of heated milk [50]. They are mainly derived from methionine degradation [51]. The SPME method (Table 2) detected dimethyl disulfide and dimethyl trisulfide in the 75 °C HTST and DSI-IUHT samples, which was consistent with the research of Hougaard et al. [52]. The species content between the samples exposed to DSI-IUHT and HTST showed more similarities than IND-UHT. The greater sterilization intensity of IND-UHT may cause non-enzymatic catalytic reactions, oxidation reactions, etc., which may lead to the degradation of the sulfur compounds in the milk.

Ketones are mainly derived from the β-oxidation reaction of saturated fatty acids in milk fat [53], yielding distinctive flavor characteristics, low flavor thresholds, and typical volatile flavor substances [54]. Only the SPME method could detect the presence of 2-heptanone and 2-nonanone, possibly due to the excellent ability of this technique to extract small and large non-polar molecules [55]. As shown in Table 2, the content of 2-heptanone and 2-nonanone increased at a higher temperature. The 2-heptanone level was 0.24 ± 0.19 μg/kg at 85 °C HTST. Although 2-nonanone was not detected in the HTST milk, it was present in the IND-UHT and DSI-IUHT samples. Therefore, 2-heptanone is a thermally induced compound formed via β-ketoacid decarboxylation and fatty acid β-oxidation, followed by decarboxylation [56]. These two ketones represent the volatile flavor compounds in milk and are responsible for its fragrance.

#### 2.2.2. Principal Component Analysis (PCA) of the Milk Samples

PCA was used to visualize the changes in the flavor compound concentrations in the five milk samples (Figure 5 and Figure 6) and reduce the dimensions of the data containing multiple variables on the premise of retaining the original information as much as possible [57].

The total variance was defined by the first two principal components (PC1 and PC2). The total variance (72.5%) in Figure 5 and the total variance (77.1%) in Figure 6 accurately reflect the types and specific roles of the volatile flavor substances. The IND-UHT milk was distinctively different from the other samples. IND-UHT is associated with aldehydes, including benzaldehyde and nonanal, explaining the fresh taste of IND-UHT milk with a distinctive burnt, bitter almond flavor [58]. IND-UHT milk is also associated with ketones, such as 2-nonanone, which may affect the milk’s flavor. Acids such as long-chain methylhexadecanoate, hexadecanoic acid, and decanoic acid are associated with IND-UHT and may be products of glyceric acid hydrolysis or degradation products from lactose. The results demonstrated that the milder DSI-IUHT flavor displayed more similarities to HTST and raw milk.

#### 2.2.3. Correlation between WPD and Flavor Substances

Protein in the food matrix itself has little to no flavor, but it can influence food flavor by binding and capturing flavor components. WP, consisting of nearly 80% β-lactoglobulin, is crucial for flavor–protein binding. β-lactoglobulin binds reversibly to flavor substances, mainly via hydrophobic, hydrogen, and non-covalent bonds, and possesses at least two binding sites. WPD may reduce flavor binding. Figure 7 shows the correlation between WPD and seven flavor substances.

The whey protein denaturation rate was positively correlated with ketone (r = 0.88, *p <* 0.05), acid (r = 0.80, *p <* 0.05), and ester (r = 0.80, *p <* 0.05) VCs and negatively correlated with alcohol (r = −0.70, *p <* 0.05), heterocycle (r = −0.57, *p <* 0.05), sulfur (r = −0.37, *p <* 0.05), and aldehyde (r = −0.28, *p <* 0.05) VCs. When exposed to heat treatment, the denaturation and decomposition of proteins produce amino acids, which are precursors of the metabolic reactions between ketones and acid catabolism, increasing the corresponding flavor substance content. Fatty acids are generally irreversibly bound to proteins via electrostatic interactions, while ketone volatiles covalently and irreversibly bind proteins by forming Schiff bases with lysine terminal amino groups [11]. Although lipids interact with proteins in a reversible hydrophobic manner, alcohols act as precursors of esters, and the thermal UWP decreases the degree of protein binding, while several alcohols esterify and hydrolyze short-chain fatty alcohols in the presence of endogenous esterases to obtain esters. Therefore, WPD is positively correlated with ketones, acids, and esters. Moreover, alcohols are reversibly bound to proteins via non-covalent interactions (hydrogen bonds) [16]. When WP is denatured, the degree of protein–alcohol binding decreases, resulting in a negative correlation between WPD and alcohol compounds. One study [59] found that the binding between WP and heptanol increased between pH 4.66 and pH 6.89. Aldehydes are usually precursors of acids and can covalently bind proteins by forming Shiff bases with amino acids. Using UPLC-TOF-MS, Anantharamkrishnan et al. [15] showed that most of the aldehydes, some of the sulfur compounds, diacetyl, and isothiocyanates in β-lactoglobulin can form covalent bonds with the flavor compounds in milk. Weel et al. [60] found that the lengths of these aldehyde carbon chains had a weak effect on retention. WPD releases sulfur-containing compounds that change flavor components and interact with proteins via irreversible binding. However, this is significantly affected by pH, with optimal binding occurring at pH 8–9 [61]. Therefore, the relevance of aldehydes and sulfur to WPD seems to be less pronounced than other flavor compounds. A close relationship between whey denaturation and VCs in cow’s milk cannot be stated at this time, as this correlation is most likely caused by the intensity of heat exposure.

## 3. Materials and Methods

### 3.1. Samples

To ensure the accuracy of the experiment, samples were collected from individuals of the same origin, pasture, and age. Raw bovine milk was provided by Junlebao Dairy Group Co., Ltd. (Shijiazhuang, China). Raw milk (200 L) was standardized to 3.9% fat milk before heat-treatment. The homogenized samples were divided equally into 4 equal portions of 50 L each and subjected to different levels of heat treatment. A two-stage homogenizer was used for the heat treatment process; the first stage’s pressure and the second stage’s pressure were 16 ± 2 MPa and 4 ± 2 MPa, respectively.

For HTST (75 °C HTST and 85 °C HTST) treatment, raw milk samples were preheated at 50 °C, homogenized, and pasteurized at 75 °C for 15 s or 85 °C for 15 s. The sample were then cooled and packed. Compared with the HTST process, the IND-UHT and DSI-IUHT samples were preheated (75 °C, 15 s) and accompanied by a secondary homogenization. Then, IND-UHT (143 °C, 3–4 s) and DSI-IUHT (143 °C, 1–2 s) were performed. Moreover, the steam used in the DSI-IUHT process was removed by flash evaporation in a vacuum cylinder at 80 ℃ and finally indirectly cooled to packing temperature. The milk samples were subjected to heat treatment, packaged in aseptic packaging, and were stored in a refrigerator at about 4 °C and equilibrated at room temperature for 20 min before experimental analysis. The whole process was performed in triplicate.

### 3.2. Particle Size Determination

The particle size and distribution of the milk samples were measured using a laser diffraction particle size analyzer (SALD-2300, Shimadzu Corporation, Kyoto, Japan) at ambient temperature. The refractive indices were set to 1.46 for the particles and 1.33 for the dispersed phase, while the accuracy was 0.001 [62]. The volume-weighted mean diameter (D[4,3]) was determined in triplicate and used to characterize the sizes of the particle.

### 3.3. Physical Stability Determination

The physical stability of the milk samples were measured in triplicate utilizing a LUMiSizer (LUM GmbH, Berlin, Germany) [63]. The instrumental parameters used for the measurement were as follows: volume, 0.4 mL of dispersion; 2500 r/min; time Exp, 7620 s; time interval, 30 s; temperature, 25 °C.

### 3.4. Apparent Viscosity Determination

The milk samples were measured in triplicate using a Brookfield DVIII viscometer (Brookfield Engineering Laboratories, Middleboro, MA, USA), an instrument used to measure the viscosity of a fluid. Optimization was based on methods described by Gruppi et al. [64] and Li et al. [65]. The measurements were performed at an invariable shear rate and ambient temperature using an SC4-18 rotor model, a rotational spindle speed of 200 r/min, and a total time of 4 min. Torque control was ≥10%.

### 3.5. Microrheological Behavior

The microrheological analyses of the milk samples were performed by a microrheological lab (Rheolaser Master, Formulation Inc., Toulouse, France). The test parameters were measured at 25 °C for 4 h. The Brownian motion of the milk droplets was measured as the MSD versus time. The EI and MVI parameters of the samples were obtained using the RheoSoft Master 1.4.0.0.

### 3.6. WPD Determination

#### 3.6.1. Sample Treatment and Denaturation Determination

The UWP was extracted by adding excess NaCl to the sample solution for final saturation. The surface charge of the protein was neutralized, destroying the hydration film on the surface to achieve casein precipitation. After a water bath, the solution was filtered while still hot, and the supernatant was collected. The NPN was extracted by mixing 15% TCA with filtered whey at a 4:1 ratio for WP precipitation [66]. The solution was left to stand, after which it was filtered to obtain the remaining NPN-containing components (amino acids and urea).

The total nitrogen concentration was used to analyze the TP, UWP, and NPN according to a method described by Dumas using an automatic Kjeldahl nitrogen analyzer (Kjeltec 8400, Foss, Copenhagen, Denmark). The analytical measurements were performed in triplicate. Native WP percentage in the raw milk samples was calculated using Equation (1):(1)$$\mathrm{Native}\;\mathrm{WP}\;\left(\%\right)=\frac{\left[\mathrm{UWP}\right]-\left[\mathrm{NPN}\right]}{\left[\mathrm{TP}\right]-\left[\mathrm{NPN}\right]}$$
where Native WP represents native protein percentage, UWP is the undenatured whey proteins, TP is the total protein, and NPN is the non-protein nitrogen.

The WPD rate in the heat-treated samples was calculated using Equation (2):(2)$$\mathrm{WPD}\;\mathrm{rate}\;\left(\%\right)=1-\left(\frac{\left[\mathrm{UWP}\right]-\left[\mathrm{NPN}\right]}{\left[\mathrm{TP}\right]-\left[\mathrm{NPN}\right]}\right)\;/\;[\mathrm{Native}\;\mathrm{WP}]$$
where WPD rate represents the degree of whey protein denaturation under heat treatment, UWP is the undenatured whey proteins, TP is the total protein, and NPN is the non-protein nitrogen.

#### 3.6.2. Sodium Dodecyl Sulphate–Polyacrylamide Gel Electrophoresis (SDS-PAGE)

To observe the UWP molecules, SDS-PAGE was performed using a method described by Genene et al. [67] and Laemmli [68], with some modifications. Based on the UWP concentrations obtained using the automated Kjeldahl nitrogen analyzer, the samples were mixed with 2x loading buffer (containing DTT) at a ratio of 3:2, boiled in water for 5 min, and centrifuged for 2 min. Finally, 15 μL of the mixture was loaded. Tris/glycine/SDS was used as running buffer. The 12% SDS-PAGE gel was prepared according to the manufacturer’s instructions (Solarbio, Beijing, China), and the protein standard (Solarbio, Beijing, China) had a molecular weight range of 0–180 kDa. After electrophoresis, the gel was stained with Coomassie brilliant blue staining solution (Brilliant Blue R, R250, Macklin) in a 60 °C water bath for 10 min, then de-stained with de-staining solution (ethanol/glacial acetic acid/H_2_O) for approximately 4 h. An image of the gel and the band intensity were obtained through a Molecular Imager^®^ Gel DocTM xr+ (Bio-Rad, Hercules, CA, USA) with Image Lab 3.0 TM Software.

### 3.7. Analysis of VCs

#### 3.7.1. SPME of the Volatile Components

Each milk sample was mixed thoroughly before extraction to ensure accuracy. A 10 g milk sample and 1 g NaCl were transferred into a 40 mL headspace bottle with a silicon septum. Next, 1 μL of dimethyl triheptanone was added to the headspace bottle as an internal standard solution. The mixture was incubated in a thermostatic water bath for 20 min at 45 °C. The 1 cm, 50/30 μm divinylbenzene/carboxen/polydimethylsiloxane fiber extraction head was exposed to the headspace above the milk sample for 30 min to allow the adsorption of volatiles. Then, the extraction head with VCs was transferred to the relevant port for gas chromatography–mass spectrometry (GC-MS). The VCs were desorbed for 5 min at 250 °C for analysis. The analytical measurements were performed in triplicate.

#### 3.7.2. SAFE of the Volatile Components

The SAFE process was performed according to a technique delineated by Havemose et al. [44], with several adjustments. The volatiles in the dichloromethane solvents were isolated. The SAFE equipment was evacuated and operated in vacuum mode at a constant voltage of 6–10 mbar while the circulating water bath was maintained at 50 °C. Next, 200 mL milk was mixed with 200 μL dimethyl triheptanone (0.816 mg/mL) internal standard and poured into the SAFE equipment to begin extraction. The extracted solution was transferred to a 250 mL dispenser funnel. Dichloromethane was added for extraction, and the organic phase was collected in a 100 mL round-bottomed flask, which was placed in a 50 °C thermostatic water bath. After distillation and concentration with a Vigreux column, the nitrogen was blown to 200 μL, after which 1 μL was extracted for GC-MS analysis. The analytical measurements were performed in triplicate.

#### 3.7.3. GC-MS Analysis

All injections occurred on an Agilent 7890 B GC with 5977 A MSD (Agilent Technologies Inc., Santa Clara, CA, USA) with a DB-WAX (60 m length × 0.25 mm i.d. × 0.25 μm film thickness; Agilent Technologies Inc., Santa Clara, CA, USA) column. The GC-MS analytical conditions were adapted from a method described by Wang et al. [69] with helium as the column carrier gas. The analytical process was performed at a constant flow rate of 1.2 mL/min in splitless mode. The GC column temperature was initially set at 40 °C with a solvent delay of 8 min, which was gradually increased at 7 °C/min to 75 °C and 2 °C/min to 150 °C, followed by a final increase at 5 °C/min to 230 °C, where it was maintained for 2 min. The inlet, ion source, and MS quadrupole detector temperatures were 250 °C, 230 °C, and 150 °C, respectively. The MS analysis was operated in electron impact mode at an electron ionization energy of 70 eV. High-resolution mass spectroscopy was performed in full scan mode in the mass range of 35–350 m/z. The C_7_–C_24_ normal alkanes were measured under the same conditions and used for calculating the retention indices (RI) [70].

### 3.8. Statistical Analysis

The results are expressed as means ± SD, one-way analysis of variance (ANOVA) by Duncan’s test was performed on the data using SPSS Statistics 17.0. Statistical significance was indicated by *p <* 0.05. SIMCA 14.1 software was employed to draw the PCA for VCs. Additionally, the correlation analysis between VCs and WPD rate was performed using OriginPro 2021 64-bit software (OriginPro Lab Corp., Northampton, MA, USA).

## 4. Conclusions

The current results suggest that the DSI-IUHT-treated milk had a 27.52% lower whey denaturation rate and displayed more similarities to raw and HTST milk in terms of odor compared to IND-UHT without changing the physical stability, mean particle size, and particle size distribution of the whey system. The apparent viscosity of the DSI-IUHT milk is significantly higher than the other samples (*p <* 0.05) and is consistent with the microrheological results. This may be related to the uniqueness of the DSI-IUHT heat treatment method, which causes proteins to cross-link and changes the droplet–droplet aggregation state via electrostatic interaction. In conclusion, these present findings suggest that DSI-IUHT is a useful strategy for improving the flavor properties of traditional UHT. It provides a theoretical basis for the production of milk with a long shelf life and good flavor in developing countries.

Concurrently, the relationship between the WPD rate and the main odor components is analyzed separately and is positively correlated with ketones, acids, and esters and negatively associated with alcohols, heterocycles, sulfur, and aldehydes. A close relationship between whey denaturation and VCs in cow’s milk cannot be stated at this time, as this correlation is most likely caused by the intensity of heat exposure. Further research is needed to increase our knowledge regarding the mechanisms of WPD and flavor.

## Figures and Tables

**Figure 1 molecules-28-03543-f001:**
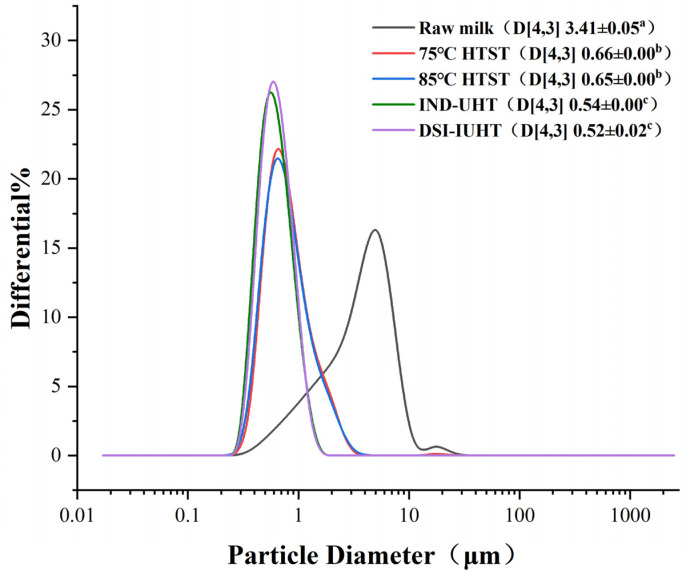
Particle sizes and distribution of the milk samples. The volume-weighted mean diameter (D[4,3]) used to characterize particle sizes. The different superscript letters of particle sizes indicate significant differences (*p* < 0.05) (75 °C HTST, 75 °C 15 s high-temperature short-time pasteurization; 85 °C HTST, 85 °C 15 s high-temperature short-time pasteurization; IND-UHT, 143 °C 3–4 s indirect ultra-high temperature; DSI-IUHT, 143 °C 1–2 s direct steam injection instantaneous ultra-high temperature).

**Figure 2 molecules-28-03543-f002:**
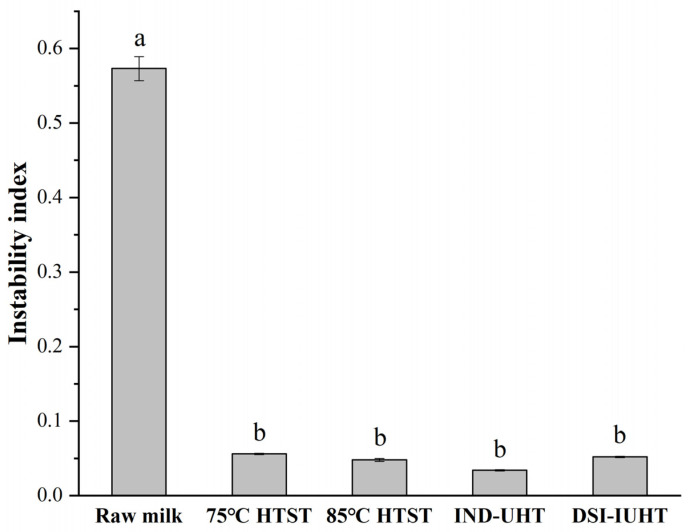
Instability index of the milk determined via a LUMISizer after heating at different temperature/time combinations. The different letters of the instability index indicate significant differences (*p <* 0.05).

**Figure 3 molecules-28-03543-f003:**
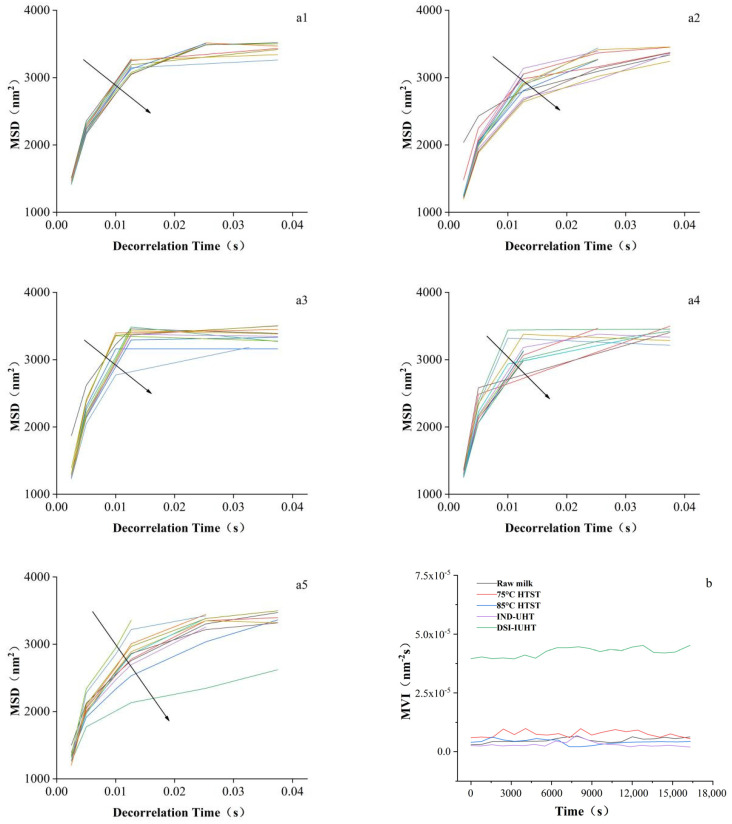
Microrheological test of the milk using a microrheolaser. Typical examples of the MSD vs. the time curves of the raw milk (**a1**), 75 °C HTST (**a2**), 85 °C HTST (**a3**), IND-UHT (**a4**), and DSI-IUHT (**a5**). The arrow represents the direction of the curve. The influence of different heat treatment methods on the MVI (**b**) and EI (**c**).

**Figure 4 molecules-28-03543-f004:**
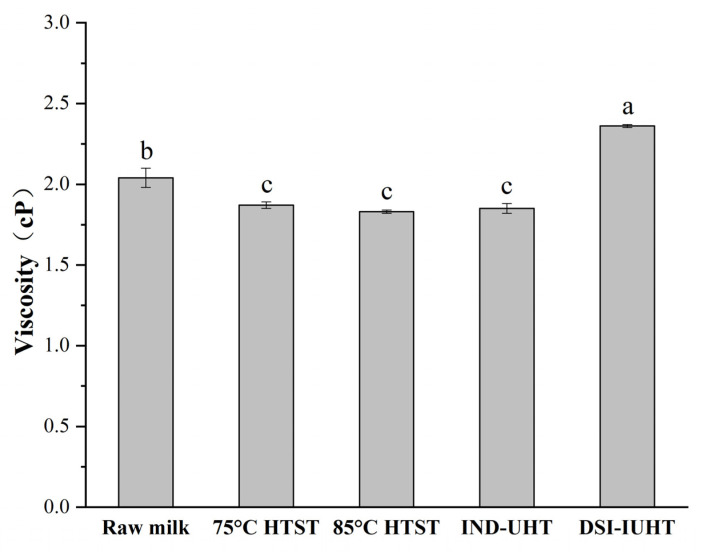
Viscosity of the milk samples at constant shear rate levels. The different letters of viscosity indicate significant differences (*p <* 0.05).

**Figure 5 molecules-28-03543-f005:**
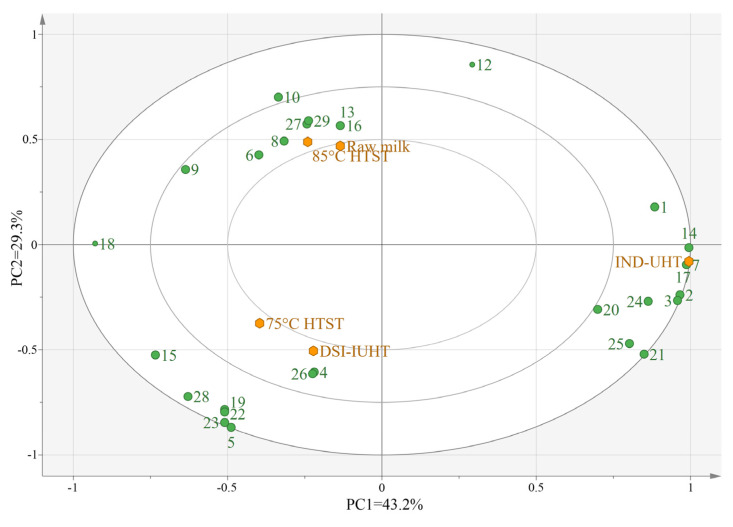
Principal component analysis (PCA) of the five milk samples subjected to SPME. The orange plots represent the five milk samples. (The numbers correspond to VCs is shown in Table 2.)

**Figure 6 molecules-28-03543-f006:**
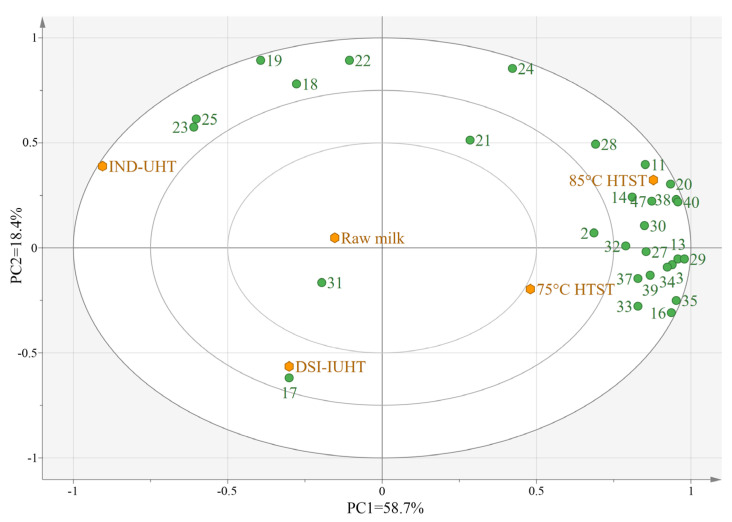
PCA of the five milk samples subjected to SAFE. The orange plots represent the five milk samples. (The numbers correspond to VCs is shown in Table 3.)

**Figure 7 molecules-28-03543-f007:**
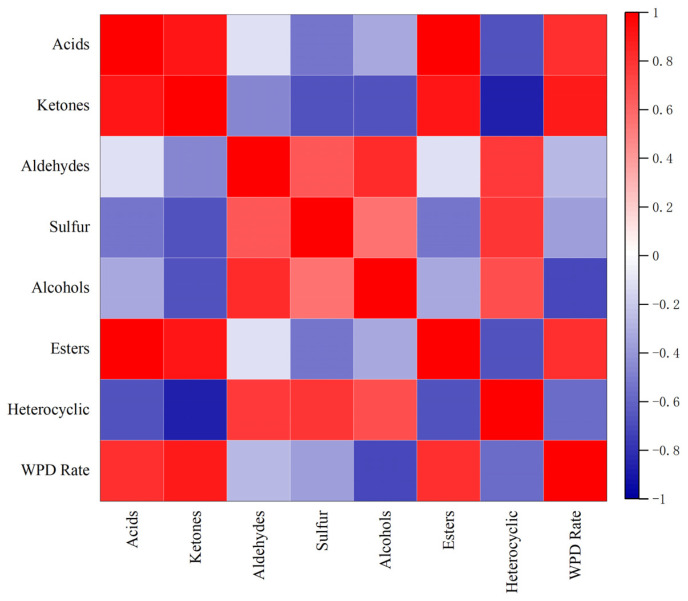
Correlation between WPD and seven flavor substances extracted by different methods (WPD Rate = Whey protein denaturation rate).

**Table 1 molecules-28-03543-t001:** Percentages of the total protein (TP), undenatured whey protein (UWP), non-protein nitrogen (NPN), and whey protein denaturation (WPD) rate for raw milk and the different heat-treated milk samples.

	Raw Milk	75 °C HTST	85 °C HTST	IND-UHT	DSI-IUHT
TP%	3.465 ± 0.009 ^a^	3.550 ± 0.164 ^a^	3.613 ± 0.094 ^a^	3.618 ± 0.081 ^a^	3.595 ± 0.023 ^a^
UWP%	0.716 ± 0.005 ^a^	0.570 ± 0.003 ^b^	0.540 ± 0.006 ^c^	0.242 ± 0.005 ^e^	0.435 ± 0.001 ^d^
NPN%	0.029 ± 0.000 ^a^	0.028 ± 0.000 ^a^	0.029 ± 0.000 ^a^	0.031 ± 0.000 ^a^	0.029 ± 0.000 ^a^
WPD rate%	-	23.06 ± 2.27 ^a^	28.64 ± 1.26 ^b^	70.59 ± 0.98 ^c^	43.07 ± 0.51 ^d^

Values are expressed as means ± SD, and the different superscript letters (within a row) indicate significant differences (*p <* 0.05).

**Table 2 molecules-28-03543-t002:** Solid-phase microextraction (SPME) analysis of the volatile compound (VC) contents in the different milk samples.

No	Compound	RI	CAS ^1^	Molecule Formula	Content (μg/kg)					Identification ^2^
					Raw Milk	75 °C HTST	85 °C HTST	IND-UHT	DSI-IUHT	
Aldehyde Compounds										
1	Nonanal	1382	124-19-6	C_9_H_18_O	ND	ND	2.59 ± 0.23	5.58 ± 0.34	ND	MS, RI
Total aldehydes					ND	ND	2.59	5.58	ND	
Ketone compounds										
2	2-Heptanone	1182	110-43-0	C_7_H_14_O	ND	ND	0.24 ± 0.19	54.41 ± 2.80	12.40 ± 0.96	MS, RI
3	2-Nonanone	1390	821-55-6	C_9_H_18_O	ND	ND	ND	25.16 ± 0.22	6.62 ± 0.78	MS, RI
Total ketones					ND	ND	0.24	79.57	19.02	
Alcohol compounds										
4	3-Hexanol	1232	623-37-0	C_6_H_14_O	ND	ND	ND	ND	9.59 ± 1.55	MS, RI
5	2-Ethyl-1-hexanol	1484	104-76-7	C_8_H_18_O	ND	0.88 ± 0.13	ND	ND	1.01 ± 0.11	MS, RI
6	1-Octanol	1606	111-87-5	C_8_H_18_O	ND	0.71 ± 0.05	1.84 ± 0.14	ND	ND	MS, RI
7	1-Dodecanol	1953	112-53-8	C_12_H_26_O	ND	ND	ND	10.62 ± 0.26	ND	MS, RI
Total alcohols					ND	1.59	1.84	10.62	10.6	
Acids compounds										
8	Formic acid	1438	64-18-6	CH_2_O_2_	ND	3.07 ± 0.39	33.46 ± 2.63	ND	4.32 ± 0.27	MS, RI
9	Acetic acid	1449	64-19-7	C_2_H_4_O_2_	3.73 ± 0.18	4.90 ± 0.61	13.50 ± 1.51	ND	6.42 ± 1.04	MS, RI
10	Butanoic acid	1637	107-92-6	C_4_H_8_O_2_	1.98 ± 0.25	0.49 ± 0.04	0.63 ± 0.03	ND	ND	MS, RI
11	Hexanoic acid	1846	142-62-1	C_6_H_12_O_2_	5..44 ± 0.18	0.85 ± 0.08	1.97 ± 0.08	0.86 ± 0.03	ND	MS, RI
12	Octanoic acid	2050	124-07-2	C_8_H_16_O_2_	8.87 ± 0.82	ND	4.37 ± 0.23	4.74 ± 0.23	ND	MS, RI
13	Nonanoic acid	2171	112-05-0	C_9_H_18_O_2_	4.12 ± 0.24	ND	ND	ND	ND	MS, RI
14	Decanoic acid	2279	334-48-5	C_10_H_20_O_2_	15.80 ± 1.05	ND	3.97 ± 0.35	112.83 ± 2.18	3.16 ± 0.09	MS, RI
15	Tetradecanoic acid	2716	544-63-8	C_14_H_28_O_2_	1.59 ± 0.10	3.07 ± 0.36	0.54 ± 0.01	ND	2.08 ± 0.25	MS, RI
16	Hexadecanoic acid	2928	57-10-3	C_16_H_32_O_2_	13.64 ± 2.04	ND	ND	ND	ND	MS, RI
Total acids					49.73	11.53	56.47	117.57	15.98	
Esters compounds										
17	Hexadecanoic acid, methyl ester	2271	112-39-0	C_17_H_34_O_2_	ND	ND	ND	9.37 ± 0.16	ND	MS, RI
Total esters					0	0	0	9.37	0	
Sulfurics components										
18	Dimethyl sulfide	1120	75-18-3	C_2_H_6_S	34.93 ± 0.94	53.91 ± 5.00	64.92 ± 8.19	ND	54.85 ± 3.88	MS, RI
19	Dimethyl disulfide	1128	624-92-0	C_2_H_6_S_2_	ND	3.64 ± 0.26	ND	ND	2.11 ± 0.31	MS, RI
20	Dimethyl sulfone	1887	67-71-0	C_2_H_6_O_2_S	ND	0.15 ± 0.02	0.31 ± 0.02	0.52 ± 0.08	0.33 ± 0.05	MS, RI
Total sulfurics					34.93	57.7	65.23	0.52	57.29	
Aromatic heterocyclic compounds										
21	Methyl-benzene	1042	108-88-3	C_7_H_8_	36.34 ± 0.42	54.06 ± 8.92	37.37 ± 6.12	101.47 ± 5.81	64.91 ± 7.20	MS, RI
22	Styrene	1254	100-42-5	C_8_H_8_	0.60 ± 0.04	7.21 ± 0.35	0.68 ± 0.10	0.66 ± 0.12	4.62 ± 0.11	MS, RI
23	P-xylene	1119	106-42-3	C_8_H_10_	ND	3.44 ± 0.56	ND	ND	2.83 ± 0.35	MS, RI
24	Limonene	1200	138-86-3	C_10_H_16_	ND	1.75 ± 0.11	ND	4.80 ± 0.35	ND	MS, RI
25	Benzaldehyde	1529	100-52-7	C_7_H_6_O	ND	ND	ND	4.15 ± 0.21	2.72 ± 0.30	MS, RI
26	4-Ethyl-benzaldehyde	1730	4748-78-1	C_9_H_10_O	ND	0.33 ± 0.02	ND	ND	25.83 ± 4.01	MS, RI
27	2-Furancarboxaldehyde	1550	98-01-1	C_5_H_4_O_2_	ND	0.95 ± 0.05	32.33 ± 2.72	0.46 ± 0.02	0.43 ± 0.03	MS, RI
28	Acetophenone	1697	98-86-2	C_8_H_8_O	ND	61.16 ± 5.49	18.72 ± 1.08	ND	45.83 ± 4.39	MS, RI
29	Maltol	2031	118-71-8	C_6_H_6_O_3_	ND	ND	10.15 ± 0.77	ND	ND	MS, RI
Total aromatic heterocyclic					36.94	128.9	99.25	111.54	147.17	

Values are expressed as means ± SD. ^1^ CAS = Chemical Abstracts Service number. ^2^ MS = mass spectra; RI = retention index. ND, not detected.

**Table 3 molecules-28-03543-t003:** Solvent-assisted flavor evaporation (SAFE) analysis of the VC contents in the different milk samples.

No	Compound	RI	CAS ^1^	Molecule Formula	Content (μg/kg)					Identification ^2^
					Raw Milk	75 °C HTST	85 °C HTST	IND-UHT	DSI-IUHT	
Aldehyde Compounds										
1	Pentanal	1037	110-62-3	C_5_H_10_O	ND	ND	ND	79.74 ± 3.22	ND	MS, RI
2	Hexanal	1035	66-25-1	C_6_H_12_O	161.20 ± 1.13	139.49 ± 11.35	100.95 ± 7.58	21.26 ± 2.89	61.27 ± 0.73	MS, RI
3	Nonanal	1382	124-19-6	C_9_H_18_O	620.31 ± 33.69	901.24 ± 19.01	849.72 ± 41.09	29.77 ± 4.92	329.09 ± 8.06	MS, RI
4	Decanal	1485	112-31-2	C_10_H_20_O	ND	116.38 ± 5.58	ND	ND	ND	MS, RI
5	Dodecanal	1711	112-54-9	C_12_H_24_O	ND	ND	699.85 ± 16.02	ND	ND	MS, RI
6	Tetradecanal	2229	124-25-4	C_14_H_28_O	ND	ND	660.78 ± 94.73	ND	ND	MS, RI
7	(Z)-6-Nonenal	1459	2277-19-2	C_9_H_16_O	ND	ND	ND	421.69 ± 46.15	ND	MS, RI
Total aldehydes					781.51	1118.57	2349.84	552.4576	390.36	
Alcohol compounds										
8	2-Methyl-3-pentanol	1167	565-67-3	C_6_H_14_O	ND	ND	ND	70.57 ± 9.31	ND	MS, RI
9	(Z)-3-Hexen-1-ol	1340	928-96-1	C_6_H_12_O	ND	ND	ND	474.71 ± 36.64	ND	MS, RI
10	(Z)-4-Hexen-1-ol	1407	928-91-6	C_6_H_12_O	ND	ND	ND	24.83 ± 2.96	ND	MS, RI
11	2-Ethyl-1-hexanol	1484	104-76-7	C_8_H_18_O	ND	60.55 ± 2.05	149.54 ± 15.80	8.86 ± 1.02	ND	MS, RI
12	3-Methyl-2-hexanol	1331	2313-65-7	C_7_H_16_O	101.88 ± 3.92	ND	ND	ND	ND	MS, RI
13	1-Dodecanol	1953	112-53-8	C_12_H_26_O	451.93 ± 3.93	665.08 ± 31.17	1050.33 ± 49.34	123.79 ± 16.75	544.24 ± 2.88	MS, RI
14	1-Tetradecanol	2200	112-72-1	C_14_H_30_O	660.65 ± 45.34	626.46 ± 40.78	714.11 ± 86.05	ND	ND	MS, RI
15	1-Hexadecanol	2400	36653-82-4	C_16_H_34_O	365.00 ± 51.16	ND	ND	ND	ND	MS, RI
Total alcohols					1579.46	1352.09	1913.98	702.75432	544.24	
Acids compounds										
16	Acetic acid	1449	64-19-7	C_2_H_4_O_2_	273.94 ± 22.12	391.62 ± 20.56	479.38 ± 25.45	ND	317.44 ± 4.97	MS, RI
17	Propanoic acid	1526	79-09-4	C_3_H_6_O_2_	55.18 ± 8.45	ND	ND	ND	57.75 ± 5.40	MS, RI
18	Butanoic acid	1637	107-92-6	C_4_H_8_O_2_	93.92 ± 9.13	ND	92.83 ± 3.31	102.03 ± 5.64	33.00 ± 1.54	MS, RI
19	Hexanoic acid	1846	142-62-1	C_6_H_12_O_2_	770.46 ± 90.07	381.85 ± 15.96	677.05 ± 38.07	957.84 ± 65.42	303.89 ± 11.12	MS, RI
20	Heptanoic acid	1918	111-14-8	C_7_H_14_O_2_	126.97 ± 14.53	183.48 ± 2.55	262.58 ± 23.76	ND	ND	MS, RI
21	Octanoic acid	2050	124-07-2	C_8_H_16_O_2_	6071.59 ± 372.34	2125.93 ± 103.15	4496.77 ± 339.10	2347.32 ± 169.45	1081.68 ± 78.52	MS, RI
22	Nonanoic acid	2171	112-05-0	C_9_H_18_O_2_	1041.09 ± 64.46	1398.84 ± 59.43	1829.44 ± 55.52	2410.64 ± 231.67	512.19 ± 67.26	MS, RI
23	Decanoic acid	2279	334-48-5	C_10_H_20_O_2_	3156.02 ± 70.91	1618.49 ± 40.82	1456.28 ± 125.20	3174.16 ± 162.65	1169.86 ± 31.19	MS, RI
24	Tetradecanoic acid	2716	544-63-8	C_14_H_28_O_2_	1560.06 ± 29.90	2161.64 ± 131.16	3006.48 ± 201.21	2415.36 ± 91.68	584.99 ± 23.58	MS, RI
25	Hexadecanoic acid	2928	57-10-3	C_16_H_32_O_2_	15,609.72 ± 896.56	16,850.52 ± 4304.50	34,828.35 ± 2413.06	84,559.69 ± 5620.35	25,243.76 ± 1286.06	MS, RI
26	9-Decenoic acid	2356	14436-32-9	C_10_H_18_O_2_	460.54 ± 9.51	ND	ND	ND	ND	MS, RI
Total acids					29,219.39	25,112.35	47,129.21	95,967.046	29,304.6	
Esters compounds										
27	Acetic acid, butyl ester	887	123-86-4	C_6_H_12_O_2_	ND	144.03 ± 0.02	295.78 ± 18.68	ND	125.51 ± 4.43	MS, RI
28	Methyl tetradecanoate	2066	124-10-7	C_15_H_30_O_2_	33.03 ± 4.06	ND	110.77 ± 20.87	ND	ND	MS, RI
29	Isopropyl myristate	2063	110-27-0	C_17_H_34_O_2_	167.62 ± 17.98	376.16 ± 27.13	384.50 ± 53.44	29.31 ± 3.94	147.05 ± 9.25	MS, RI
30	Hexadecanoic acid, methyl ester	2281	112-39-0	C_17_H_34_O_2_	2570.46 ± 33.78	6715.59 ± 404.62	5090.39 ± 117.87	702.90 ± 44.01	989.49 ± 86.48	MS, RI
31	Hexadecanoic acid, ethyl ester	2288	628-97-7	C_18_H_36_O_2_	1346.74 ± 10.11	ND	ND	ND	377.65 ± 20.43	MS, RI
32	Octadecanoic acid, methyl ester	2445	112-61-8	C_19_H_38_O_2_	1277.59 ± 72.60	3420.89 ± 66.30	1955.53 ± 60.09	ND	ND	MS, RI
33	Dibutyl phthalate	2393	84-74-2	C_16_H_22_O_4_	ND	14,894.02 ± 473.02	12,460.73 ± 764.53	ND	7270.73 ± 114.18	MS, RI
Total esters					5395.44	25,550.67	20,297.67	732.21	8910.429	
Sulfurics components										
34	Dimethyl sulfide	760	75-18-3	C_2_H_6_S	157.11 ± 1.14	217.85 ± 3.87	409.93 ± 11.77	ND	210.18 ± 3.60	MS, RI
35	Dimethyl sulfone	1887	67-71-0	C_2_H_6_O_2_S	495.56 ± 2.08	629.96 ± 38.29	760.41 ± 19.10	166.31 ± 20.96	525.89 ± 4.26	MS, RI
Total sulfurics					652.67	847.81	1170.34	166.309	736.07	
Aromatic heterocyclic compounds										
36	Toluene	1042	108-88-3	C_7_H_8_	ND	ND	ND	311.89 ± 13.42	ND	MS, RI
37	Methyl-benzene	1105	108-88-3	C_7_H_8_	1071.28 ± 8.97	3029.22 ± 110.98	1816.26 ± 52.82	ND	557.92 ± 15.13	MS, RI
38	p-Xylene	1119	106-42-3	C_8_H_10_	288.45 ± 5.27	415.01 ± 15.36	833.04 ± 37.62	ND	152.99 ± 1.62	MS, RI
39	Ethylbenzene	1123	100-41-4	C_8_H_10_	59.99 ± 6.84	108.94 ± 4.14	228.55 ± 9.11	ND	129.10 ± 15.52	MS, RI
40	Limonene	1200	138-86-3	C_10_H_16_	535.53 ± 13.46	1313.19 ± 21.90	1589.56 ± 55.73	76.83 ± 5.34	132.16 ± 5.30	MS, RI
41	4-Ethyl-benzaldehyde	1730	53951-50-1	C_9_H_10_O	ND	ND	ND	ND	86.67 ± 7.67	MS, RI
42	2-Furanmethanol	1711	98-00-0	C_5_H_6_O_2_	ND	ND	ND	11.29 ± 0.45	ND	MS, RI
43	2(5H)-Furanone	1767	497-23-4	C_4_H_4_O_2_	ND	ND	ND	161.53 ± 8.62	ND	MS, RI
44	Acetophenone	1699	98-86-2	C_8_H_8_O	ND	ND	ND	ND	50.72 ± 1.07	MS, RI
45	Naphthalene	1779	91-20-3	C_10_H_8_	ND	56.85 ± 3.87	ND	ND	ND	MS, RI
46	2-Methyl-naphthalene	1891	91-57-6	C_11_H_10_	ND	96.41 ± 13.75	ND	ND	ND	MS, RI
47	Butylated Hydroxytoluene	1956	128-37-0	C_15_H_24_O	672.25 ± 2.92	480.87 ± 15.08	1282.75 ± 72.55	ND	327.57 ± 28.19	MS, RI
Total aromatic heterocyclic					2627.5	5500.49	5750.16	561.5468	1437.13	

Values are expressed as means ± SD. ^1^ CAS = Chemical Abstracts Service number. ^2^ MS = mass spectra; RI = retention index. ND, not detected.

## Data Availability

Data are contained within the article.

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
