# Peer review of "Effect of Direct Steam Injection and Instantaneous Ultra-High-Temperature (DSI-IUHT) Sterilization on the Physicochemical Quality and Volatile Flavor Components of Milk"

_molecules, 2023, doi:10.3390/molecules28083543_

Round 1
Reviewer 1 Report
Please review the publication for grammatical and spelling errors and inaccuracies. Such as:
Line 176: "whey protein denaturationl"... and for the WPD rate, although it was stated that there was a difference in the study, no statistical difference was shown in Table 1.
Author Response
Dear reviewers,
Many thanks to the reviewers for their constructive suggestions, which have helped us to improve the quality and depth of the paper in English. We have carefully studied these comments and made changes. Details are in the attachment.
Once agin, Thank you for your comments and suggestion.
Best regards.
Yours sincerely
Yan Li

Reviewer 2 Report
1. This article is helpful to developing nations, especially those who require longer transportation times yet lack complete cold-chain transportation. Unfortunately, the sterilization method mentioned in this article had already been applied, hence the article won't be considered particularly inventive.
2. Steam will be produced during the Direct Steam Injection Instantaneous Ultra-High Temperature (DSI-IUHT) Sterilization; kindly specify the improvement.
3. The results indicated that sterilization had a significant impact on milk, but they did not adequately illustrate the necessity of using this system.
4. It can be seen from the results of Figures 1, 2, and 3 that there were variations following sterilization, but the findings were not discussed in the discussion.
5. There should be more discussion of odor-active coumpounds.
Author Response

(The authors gave the same response as above.)

Reviewer 3 Report
Please see the attached file

Author Response

(The authors gave the same response as above.)
